# OpenReview forum: "Comparative analysis of black box methods for detecting evaluation awareness in LLMs"
_ICLR.cc/2026/Conference — Submitted to ICLR 2026_

### Official Review · Reviewer_m5bt · 2025-10-26

**Soundness:** 1
**Presentation:** 1
**Contribution:** 1
**Rating:** 2
**Confidence:** 5

**Summary:**

This work evaluates how llms may behave differently when it is aware of being evaluated.

**Strengths:**

Some of the models tested are new, though it is hard to justify why those models rather than others.

**Weaknesses:**

The layout, writing style, and the design of the experiments are not very professional nor scientific. The authors seemed to have tried very hard to stretch a class project or a workshop paper into 9 pages without adding substantially more results or experiments. And **the provided hugging face link contains authors name, which has violated the double-blind policy unfortunately.** The reviewer kindly recommend the authors to take each year's accepted paper as reference, and follow their layout and similar design of research presentations and experiements.

**Questions:**

Please refer to the weakness section.

---

### Official Review · Reviewer_y64m · 2025-10-27

**Soundness:** 2
**Presentation:** 1
**Contribution:** 1
**Rating:** 2
**Confidence:** 4

**Summary:**

This paper attempts to provide an empirical study to determine whether LLMs are evaluation-aware. It aims to identify which methods can more reliably probe a model’s evaluation awareness, what prompt elements cause models to realize they are being evaluated, and ultimately how to bypass the model’s evaluation-aware state to provide more objective and reliable evaluation results.
In addition to existing methods, the paper proposes two new approaches to probe evaluation awareness: User Motivation and Log Probability.

**Strengths:**

The problem addressed in this paper is important. The authors systematically review existing methods and evaluate their performance, which is helpful for the development of future methods.

**Weaknesses:**

- The paper is incomplete, poorly organized, and difficult to understand.

- The two proposed methods perform poorly and lack further optimization. For example, in the log probability method, DeepSeek tokenizes "evaluation" into "e" and "valuation", but the authors do not explore how to handle such noise.

- The paper includes very few experiments, which prevents it from providing useful conclusions. It also fails to answer the research questions this paper raises. For instance, which sections discuss how to bypass the model’s evaluation-aware functions? Which sections explore what prompt elements cause models to realize they are being evaluated? The authors do not address these questions.

**Questions:**

The paper needs significant improvement.

---

### Official Review · Reviewer_wwUy · 2025-10-29

**Soundness:** 2
**Presentation:** 2
**Contribution:** 2
**Rating:** 2
**Confidence:** 5

**Summary:**

This paper investigates evaluation awareness in LLMs—whether models can detect when they're being evaluated versus deployed. The authors use primarily prompting-based methods to  compare five black-box detection approaches on 976 conversation transcripts, and construct a taxonomy of prompt features that trigger evaluation awareness.

**Strengths:**

* Evaluation awareness is a critical issue for AI safety and alignment research

**Weaknesses:**

* Fundamentally flawed baseline wastes limited space: The log probability analysis method is acknowledged by authors as "essentially unusable", yet receives a full methods subsection, results section, and figure—space better spent on deeper analysis of viable methods
* Missing clear definition of evaluation-awareness: The paper conflates casual user testing (e.g., trying a new model) with safety-critical evaluations that determine model release or trigger gradient updates during post-training. Lacks fine-grained distinctions between capability vs. alignment evaluations as established in prior work, such as Capability and Alignment , as introduced by [1]
* Priming effects undermine construct validity: Most methods explicitly mention "evaluation" or ask models to reason about user intent, potentially creating the very awareness being measured rather than detecting naturally occurring evaluation awareness
* No human baseline for comparison: Unclear whether distinguishing evaluation from deployment prompts is even feasible for human raters, making it impossible to assess whether poor model performance reflects task difficulty or genuine inability

[1]Large Language Models Often Know When They Are Being Evaluated

**Questions:**

* Recommendation: Please consider submitting to ICLR Blogposts 2026 track instead—this exploratory analysis with negative results would make a valuable contribution to the community discussion without claiming finished research ready for the main conference.

---

### Official Review · Reviewer_EBYX · 2025-10-29

**Soundness:** 1
**Presentation:** 1
**Contribution:** 1
**Rating:** 2
**Confidence:** 3

**Summary:**

This paper investigates whether large language models can recognize when they are being evaluated (evaluation awareness). Five detection methods are tested under the same dataset and experimental conditions, and differences in model responses and behaviors are analyzed.

**Strengths:**

- The paper compares five methods (Binary, Probability, User Motivation, Reasoning Traces, Log Prob) using the same dataset and identical experimental conditions.
- The evaluation covers five models: GPT-5 Mini, Gemini 2.5 Pro, DeepSeek R1, gpt-oss-120b, and Qwen3-32B.

**Weaknesses:**

- While the topic of evaluation awareness is highly interesting, the verification methodology remains insufficient in several respects.
- Although the authors state that “Evaluation awareness requires the model to show understanding that the task itself is likely to be an assessment, test, or evaluation of some sort, not just a normal user request,” they do not provide a unified definition of what constitutes such “understanding.”  Each of the five methods implicitly adopts a different operational definition.  Because of this inconsistency, the results across the five methods are not coherent, making it difficult to determine whether the methods themselves are flawed or whether the phenomenon of evaluation awareness actually exists.
- At L452 the authors write, “It might be either because the methods used here are fundamentally flawed, or LLMs genuinely might not exhibit much evaluation awareness. It’s hard to tell.”  However, the paper offers no conclusive analysis to decide between these two explanations and provides no further investigation to identify which is correct.

**Questions:**

There appears to be considerable room for improving the overall quality of the paper (for example, by clarifying its position relative to prior research and explicitly situating the study within the broader literature).

---

### Meta-Review · Area_Chair_Nuwo · 2026-01-07

**Summary:**

The paper investigates the evaluation awareness of LLMs, i.e. can they identify if they are being evaluated. While authors unanimously agree that this is an interesting problem space, they point out multiple issues with the framework, including (1) missing well-stated definition of evaluation awareness, (2) issues with solution design, particularly with the log probability based method, amongst other issues.

**Reviewer Concerns:**

The authors did not submit any rebuttal.

**Reviewer Scores:**

The authors did not submit a rebuttal.

---

### Decision · Program_Chairs · 2026-01-26

Reject